# LEARNING HOW TO INFER PARTIAL MDPS FOR IN-CONTEXT ADAPTATION AND EXPLORATION

**Chentian Jiang** *
Informatics
University of Edinburgh
Edinburgh, UK
`chentian.jiang@ed.ac.uk`

**Nan Rosemary Ke & Hado van Hasselt**
DeepMind
London, UK

## ABSTRACT

To generalize across tasks, an agent should acquire knowledge from past tasks that facilitate adaptation and exploration in future tasks. We focus on the problem of *in-context* adaptation and exploration, where an agent only relies on context, i.e., history of states, actions and/or rewards, rather than gradient-based updates. Posterior sampling (extension of Thompson sampling) is a promising approach, but it requires Bayesian inference and dynamic programming, which often involve unknowns (e.g., a prior) and costly computations. To address these difficulties, we use a transformer to *learn* an inference process from training tasks and consider a hypothesis space of *partial models*, represented as small Markov decision processes that are cheap for dynamic programming. In our version of the Symbolic Alchemy benchmark, our method's adaptation speed and exploration-exploitation balance approach those of an exact posterior sampling oracle. We also show that even though partial models exclude relevant information from the environment, they can nevertheless lead to good policies.

## 1 INTRODUCTION

To generalize well, an agent should acquire knowledge from past tasks that helps it explore and adapt effectively in future tasks. In deep reinforcement learning (RL), there is growing interest in *in-context* generalization, where upon entering a new task, an agent's neural network weights are frozen and its policy explores and adapts based solely on real-time context, i.e., history of states, actions and/or rewards (e.g., Wang et al., 2016; Duan et al., 2016; Laskin et al., 2022).

Taking inspiration from human behavior (e.g., Speekenbrink & Konstantinidis, 2015; Schulz et al., 2015), in-context generalization can be approached through a kind of hypothesis testing: An agent can acquire prior beliefs from past tasks about how tasks tend to work, e.g., what are the typical dynamics governing interactions between objects? In an unfamiliar (but related) task, it can periodically sample one promising hypothesis under its beliefs and test it by acting optimally under that hypothesis. The resulting data, or context, would inform the agent whether its hypothesis was correct or not, as well as improve its broader posterior beliefs. By sampling and testing many hypotheses, an agent would try diverse actions to *explore* a new task and gain information to *adapt* its posterior beliefs toward how the task truly works. The agent's policy would also adapt as it acts optimally under more accurate hypotheses.

This hypothesis testing concept is neatly encapsulated by a computational framework called *posterior sampling* Strens (2000); Osband et al. (2013), which extends *Thompson sampling* Thompson (1933) from multi-armed bandits to RL. Posterior sampling, however, requires Bayesian inference and dynamic programming, which are intractable for all but toy problems and may need unknown information (e.g., a prior distribution). Even when these challenges are addressed by neural network approximations of posterior sampling (e.g., Osband et al., 2019; 2016; Azizzadenesheli & Anandkumar, 2019; Lipton et al., 2018; Fortunato et al., 2018), these methods are typically designed for training and evaluation on the same task, rather than for generalization across tasks.

---

*Work was conducted during an internship at DeepMind.

Our method for approximating posterior sampling tackles all the difficulties above: We approximate inference rather than relying on Bayesian inference, and we do so for a hypothesis space of *partial* models—represented as small Markov decision processes (MDPs)—that are cheap for dynamic programming. Furthermore, we treat the approximate inference process as generalizable knowledge that is learned from training tasks (using a neural network) and later applied to held out testing tasks.

To illustrate a partial model, imagine an empty wine glass on a table. And consider a small MDP model whose dynamics only capture that pushing the glass has a high probability of breaking the glass. This model is incomplete in the sense that it excludes other relevant information for predicting whether the glass breaks, such as the glass's position on the table. However, since wine glasses generally have a round shape, we might expect they are likely to roll off the table and break, no matter where they were on the table. By applying dynamic programming on our model, we may not find an optimal policy that accounts for all relevant information, but we can nevertheless find a decent policy that avoids pushing the wine glass and is faster to compute.

To learn an inference process for partial models, our method trains a transformer Vaswani et al. (2017) because transformers have previously demonstrated the ability to approximate Bayesian inference Müller et al. (2022); Xie et al. (2022) and generalize in-context Brown et al. (2020); Garg et al. (2022). We build on architectures that take in an entire data set—or an entire context/history in our work—rather than a single data point Kossen et al. (2021); Goyal et al. (2022); Ke et al. (2022); Müller et al. (2022). Given context, our transformer outputs a distribution over partial models, represented as small graphs. At testing time, we freeze our transformer's weights and embed it as an approximate inference machine in the posterior sampling framework.

We evaluate our method on a simplified version of the Symbolic Alchemy benchmark Wang et al. (2021). It provides a distribution of tasks for studying our generalization problem, i.e., training on some tasks (with varying transition functions) and measuring in-context adaptation and exploration performance in held out tasks. As distinct from most other benchmarks, Alchemy provides good partial representations of tasks, essentially associating a ground truth partial model with each task. This ground truth is used as a supervision signal for training our transformer, allowing us to defer the problem of how partial models should be represented (e.g., what from the environment should be included or excluded in the partial model's state space?). We instead focus on learning *how to infer* partial models from training tasks and testing whether there is merit in using partial models with the posterior sampling framework. Our method approaches the behavior of an exact posterior sampling oracle, both in terms of in-context adaptation speed and exploration-exploitation balance. We also affirm that partial models can facilitate adaptation toward near-optimal rewards.

## 2 PROBLEM SETTING

**Task distribution.** An MDP model of an environment has states $s \in \mathcal{S}$, actions $a \in \mathcal{A}(s)$, a transition probability function $p(s'|s, a)$, a reward function $r(s, a)$, a discount factor, and a starting state distribution. We tackle a problem setting with a distribution of tasks $p(k)$ (represented as MDPs), where transition and reward functions are deterministic and can vary across tasks. By learning from a set of training tasks $k_{\text{train}} \sim p(k)$, an agent should grasp the general structure of possible transition and reward functions, and generalize well to testing tasks $k_{\text{test}} \sim p(k)$. We hold out the testing tasks such that they are never encountered during training, and evaluate generalization by how well an agent adapts and explores in-context in those testing tasks. Our problem statement is similar to several meta-RL works, which we discuss in Section 7.

**Defining in-context adaptation and exploration.** We define adaptation and exploration for the purposes of our work (but do not claim these definitions cover related works). We consider adaptation to be the process of (1) inferring partial models that are more accurate descriptions of a part of the environment and (2) producing better policies as measured by reward. Importantly, such adaptation should occur *in-context*, where context refers to a history of actions and states: At testing time, models and policies should improve as a function of conditioning on longer contexts (i.e., more states and actions)—akin to conditioning on more data in Bayesian inference—as opposed to using gradient-based updates. We consider meaningful exploration to be collecting new context that is informative for inferring partial models, i.e., for disambiguating between them. As an approximation, we measure the number of unique state-actions visited in the partial model space.

## 3  LEARNING HOW TO INFER PARTIAL MODELS

Our method learns general knowledge from training tasks in the form of an *inference process over partial models*: given context (a history of states and actions), what is the distribution of partial transition functions and partial reward functions that best describes the current task?

**Partial MDP.**   Our partial MDP models describe only *a part of* the environment by mapping the full state space $\mathcal{S}$ to a smaller space $\mathcal{V}$ (many-to-one) and defining simpler transition functions and reward functions on this space. We represent partial models as graphs: states are vertices $v \in \mathcal{V}$, actions $x \in \mathcal{X}(v)$ are edge traversals, a transition function is represented as a vector of the probability of each edge existing $\mathbf{e} \in [0,1]^N$, and a reward function is represented as a vector of the probability of each vertex-edge combination having a positive (=1) reward $\mathbf{r} \in [0,1]^M$ (otherwise a vertex-edge has zero reward).

**Approximating distributions.**   Unlike the usual way of representing transition and reward functions as a neural network (e.g., Ha & Schmidhuber, 2018; Schrittwieser et al., 2020), we represent them as a vector of edge probabilities $\mathbf{e}$ and reward probabilities $\mathbf{r}$, respectively, that are the *output* of a neural network, not the network itself. Our representation allows our method to focus on learning a small graph rather than all the details in the environment's dynamics.

Our method learns an approximation of the inference process $p(\mathbf{e}, \mathbf{r}|\mathbf{C}_{1:t})$. $\mathbf{e}$ is a vector of edge probabilities in our partial model graph, $\mathbf{r}$ is a vector of reward probabilities, and $\mathbf{C}_{1:t}$ is a matrix containing the context, which is an agent's history of states and actions $(s, a)$ up to time $t$. We use a transformer $f_\theta$ to define the distribution:

$$\hat{p}_\theta(\mathbf{e}, \mathbf{r}|\mathbf{C}_{1:t}) = \prod_n \mathrm{Ber}(e_n|f_{\theta,n}(\mathbf{C}_{1:t}))$$
$$\times \prod_m \mathrm{Ber}(r_m|f_{\theta,m}(\mathbf{C}_{1:t}))$$

$e_n \in \{1, 0\}$ is an edge probability enumerated by $n$ and $r_m \in \{1, 0\}$ is a reward probability enumerated by $m$. Ber is a Bernoulli distribution whose probability parameter is our transformer's sigmoid function output for either the $n$th edge, $f_{\theta,n}(\mathbf{C}_{1:t})$, or the $m$th reward location (i.e., $m$th vertex-edge), $f_{\theta,m}(\mathbf{C}_{1:t})$. For simplicity, we have assumed independence between all Bernoulli distributions. Appendix A has details of our transformer architecture.

Our transformer also has a softmax function output that defines a categorical distribution over vertices $\hat{p}_\theta(v_t|s_t)$ (our transformer takes in the entire context, but it should learn that $v_t$ only depends on the most recent state $s_t$).

**Training.**   For each training task $k$, we assume access to supervision signals (marked with an asterisk) for a partial representation of its transition function $\mathbf{e}_k^*$ and reward function $\mathbf{r}_k^*$. For each state $s_t$ encountered within a task, we assume access to the state's corresponding vertex $v_t^*$ in the partial model space. These supervision signals give us a good representation of partial models, enabling us to instead focus on (1) learning an inference process $\hat{p}_\theta(\cdot|\mathbf{C})$ for such representations and (2) testing whether these representations are reasonable to use with posterior sampling even though they are not complete MDPs. Our transformer is trained autoregressively by minimizing the following loss:

$$\mathcal{L}(\theta) = \mathbb{E}_{p(\mathbf{C}_{1:T}, k)} \left[ \frac{1}{T} \sum_{t=1}^{T} \left[ -\ln(\hat{p}_\theta(\mathbf{e}_k^*, \mathbf{r}_k^*|\mathbf{C}_{1:t})) \right. \right.$$
$$\left. \left. - \ln(\hat{p}_\theta(v_t^*|s_t)) \right] \right]$$

where $p(\mathbf{C}_{1:T}, k)$ is a joint distribution over contexts, each with an episode-length number of time steps $T$, and tasks.

We generate offline training data that represent samples from $p(\mathbf{C}_{1:T}, k)$: for each training task $k$, we generate $M$ episode-length contexts $\mathbf{C}_{1:T}$ (i.e., history of states and actions) using a random policy.

---

**Algorithm 1** Posterior Sampling

---

**Require:** Trained transformer parameters $\theta$
  Initialize empty context matrix $\mathbf{C}$
  **for** time step $t$ in testing episode **do**
    **if** full method **then**
      Sample $\mathbf{e}, \mathbf{r} \sim \hat{p}_\theta(\mathbf{e}, \mathbf{r}|\mathbf{C})$
    **else if** ablation **then**
      $\mathbf{e} \leftarrow \mathbb{E}_{\hat{p}_\theta(\mathbf{e}|\mathbf{C})}[\mathbf{e}]; \mathbf{r} \leftarrow \mathbb{E}_{\hat{p}_\theta(\mathbf{r}|\mathbf{C})}[\mathbf{r}]$
    **end if**
    Value iteration with $\mathbf{e}$ and $\mathbf{r}$ (transition and reward functions in partial model space) produces $\pi : \mathcal{V} \to \mathcal{X}$
    $v_t \leftarrow \arg\max_{v_t} \hat{p}_\theta(v_t|s_t)$
    $x_t \leftarrow \pi(v_t)$
    Map $x_t$ to $a_t$ (action in real env); observe $s_{t+1}$
    Append $(a_t, s_{t+1})$ as a row to $\mathbf{C}$
  **end for**

---

Each context is associated with the supervision signals for the corresponding task $\{\mathbf{e}_k^*, \mathbf{r}_k^*\}$ and with the supervision signal about how each state in the context maps to a vertex $v_t^*$.

## 4   Posterior Sampling

At testing time, our transformer's weights $\theta$ are frozen. As opposed to its offline training, it now makes *online* inferences within the posterior sampling framework (Algorithm 1).

As a general framework, posterior sampling sets a prior distribution over possible models of the environment. It then samples a single model from this distribution and solves for the optimal policy of that model using dynamic programming. The resulting policy is followed for some time steps (1 time step in our work), producing data for updating the posterior distribution over models. The next model is sampled from the posterior distribution and the rest of the process is repeated.

In our work, we approximate the inference process and use *partial* models instead of complete ones. We replace the prior with $\hat{p}_\theta(\mathbf{e}, \mathbf{r}|\mathbf{C}_0)$, where $\mathbf{C}_0$ is an empty context. Given new data, i.e., states and actions, we append those to the context $\mathbf{C}_{1:t}$ and replace the posterior update with $\hat{p}_\theta(\mathbf{e}, \mathbf{r}|\mathbf{C}_{1:t})$. A deterministic partial model, i.e., $\mathbf{e}$ and $\mathbf{r}$ containing 1/0 probabilities, is then sampled and solved with dynamic programming, particularly value iteration, producing a policy $\pi$ in the partial model space.

The current vertex prediction $v_t$ from our transformer is used to query an action $x_t \leftarrow \pi(v_t)$ in the partial model space. Finally, we assume that there is a known one-to-one mapping between partial model actions $\mathcal{X}$ (edge traversals) and real actions in the environment $\mathcal{A}$. After mapping $x_t$ to $a_t$, our method executes $a_t$ in the real environment and observes the next state $s_{t+1}$. $(a_t, s_{t+1})$ are then appended to the context $\mathbf{C}_{1:t+1}$.

Unlike the original posterior sampling proposal Osband et al. (2013); Strens (2000), our method samples a model every time step rather than after a longer time, e.g., an episode. In some settings, acting under a sampled model for a longer time may produce more informative data for helping an inference machine discriminate between models Russo et al. (2020). However, in our environment (Section 5), the immediate state-action data can be highly informative for discriminating between partial models, so we prefer to update our partial model beliefs and sample at every time step. Modifying our method toward the original posterior sampling setup simply requires waiting longer before sampling.

**Ablation.** We construct an ablation that removes the sampling component in our posterior sampling procedure (Algorithm 1), thereby testing how the presence vs. absence of sampling affects adaptation and exploration. Instead of sampling, we use the expectations of $\mathbf{e}$ and $\mathbf{r}$, which are the direct outputs from our transformer (i.e., parameters for each edge's Bernoulli distribution, and parameters for each reward location's Bernoulli distribution, respectively). These expected values now represent

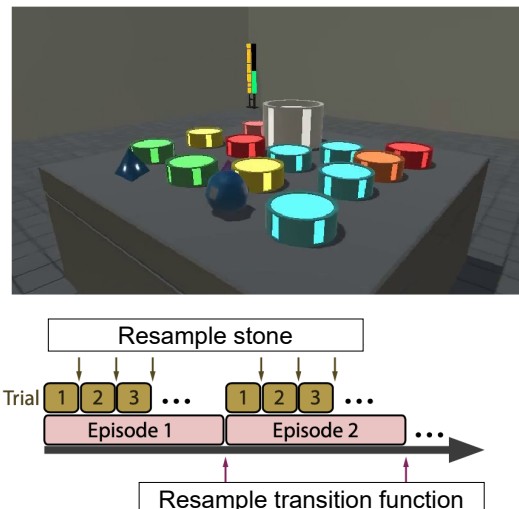

Figure 1: Alchemy environment (adapted from Figure 1 of the Alchemy paper Wang et al. (2021)). **Top:** Visualization of stones (sphere and pyramid objects), potions, and a cauldron (gray cylinder). In our work, an agent observes a *symbolic* (rather than 3D) version of Alchemy with only 1 stone and 6 potions. An agent's goal is to transform its stone into a more rewarding one by dipping the stone into potions. **Bottom:** Episode structure.

*stochastic* transition and reward functions (as opposed to deterministic ones produced by sampling): an edge traversal is successful with some 0-1 probability and a vertex-edge combination has a positive reward with some 0-1 probability.

## 5 EXPERIMENTAL SETUP

### 5.1 SIMPLIFIED SYMBOLIC ALCHEMY ENVIRONMENT

We focus on the Symbolic Alchemy benchmark Wang et al. (2021) (Figure 1 top) because it (1) provides access to partial model representations of tasks for training our transformer, and (2) can evaluate how well our method adapts and explores in held out tasks. The full Symbolic Alchemy environment is a playground for investigating a range of problems, such as learning how to act on multiple objects and learning varying mappings from latent states to perceived observations. We pick the problem of generalizing across tasks, and for simplicity, focus only on tasks with a fixed reward function and varying transition functions. We simplify Symbolic Alchemy to isolate this problem.

In our environment, an agent receives symbolic observations of 1 stone and 6 potions. It can dip the stone into a potion, which empties the potion and may transform the stone's color, shape or size. Each of the 6 potions has a unique effect targeting one feature (e.g., size) and one kind of change (i.e., increase or decrease). Different combinations of stone features are associated with different rewards (-3, -1, 1 or 15), and the agent's goal is to transform its initial stone features into more rewarding ones. Higher rewards are indicated by higher brightness values on a stone (small patches in Figure 1). *To actually receive a stone's reward, the agent must deposit its stone into the cauldron (cylindrical container in Figure 1).* It must do so by the end of a *trial*, which consists of 10 time steps (there are 20 trials in an episode). After a stone has been deposited into the cauldron, it disappears and the agent must wait until the next trial to receive a new stone with randomly sampled features.

**Partial Models.** Our environment associates each task with a partial model, where each task has a unique partial transition function. These models are represented as constrained graphs, called "cubes" (Figure 2), that are closely based on Alchemy's original partial model representations. They have 8 vertices and 12 edges, where edges can only exist along the outline of a cube plotted in a 3-dimensional space, with each dimension representing a stone feature (color, shape, size). The vertices represent the possible feature combinations for a single stone, and only one vertex is associated with a positive reward, representing the stone that gives the highest reward (15) if it is deposited into the

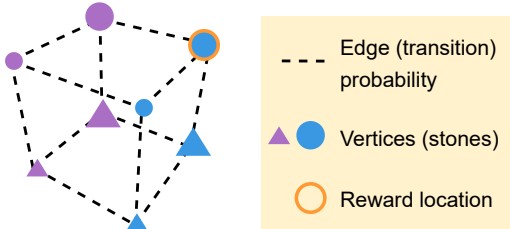

Figure 2: Partial model (also called "cube") for our simplified Alchemy environment.

cauldron. The existence of an edge represents whether it is possible to transition the stone between two vertices (i.e., sets of features). If an edge exists, then the stone can be transformed by acting on a particular potion, *assuming the potion is not empty*. Acting on a potion is represented as traversing one dimension and direction in the 3D cube space (e.g., the yellow potion makes the stone bigger, i.e., it corresponds to the increasing direction of the size dimension). In this way, the cube represents a small MDP, where vertices are states, edge traversals are actions, the vertex associated with positive reward represents a reward function, and the existence of edges represents a transition function.

The MDP represented by a cube is a *partial* model of a task because it omits information. For example, it does not capture how potions become empty after they are used (and can no longer transform the stone) or how some stones do not have the maximum reward but have small positive rewards (+1). Our method performs dynamic programming (i.e., value iteration) on this kind of partial model, which amounts to finding the shortest path from the current vertex/stone to the one with the highest reward. The resulting policy could be suboptimal; it might try empty potions, and it would not deposit low (but positive) reward stones into the cauldron (but rather get stuck with zero reward if the remaining potions cannot produce the highest reward stone). Nevertheless, we expect that cubes are useful partial models because they capture essential information about transitions in the full environment. And dynamic programming may produce policies that only need to use each potion at most once to reach the highest reward stone, thereby achieving the optimal trial reward without needing information about empty potions or low-reward stones. Therefore, we hypothesize that dynamic programming on these partial models would produce policies that often result in high rewards.

Alchemy gives researchers access to the cubes underlying its tasks, allowing us to pick a ground truth partial model, generate a task from that model, and collect trajectory data using a random policy. The true partial model and the trajectory data become the supervision signal and context, respectively, for training our transformer. Out of 109 possible cubes, we use 88 for training, 11 for validation (i.e., for choosing hyperparameters) and 10 for testing. Details of how we implement our method are in Appendix B.

**Evaluation.** Our episodes are split into 20 trials, each consisting of 10 time steps (Figure 1 bottom). Within an episode, the ground truth partial model stays the same. Across trials in the same episode, the stone is reset with a sampled set of features (equivalent to sampling a vertex on the cube), and all potions are reset to being full again. Across episodes, the presence/absence of edges in the true partial model changes, meaning the transitions governing how potions transform stones change. To obtain high rewards, an agent must generalize well to new episodes: it must explore new stone-potion transitions and adapt its policy to transition toward the most rewarding stone.

We consider a held out set of 10 Alchemy cubes in our evaluation. For each cube, we generate 20 episodes, each with a differently sampled sequence of stone resets. We evaluate our method within the span of a single episode (20 trials per episode = 200 time steps). At the beginning of each evaluation episode, both our method and ablation begin with the same trained transformer and an empty context. All metrics, explained below, are reported as averages across the 10x20=200 evaluation episodes.

We measure policy adaptation as an agent's percentage of optimal reward per trial. Raw rewards are divided by the rewards from an optimal policy produced by solving (with value iteration) the *true and complete MDP*. The optimal policy's trial reward is 15 whenever the best stone can be produced given the trial's starting stone and potions, but is 1 otherwise. We also measure model adaptation as the Mean Absolute Error (MAE) between predicted probabilities of an edge existing (i.e., the Bernoulli parameters returned by our transformer in the case of our full method or ablation) and the true

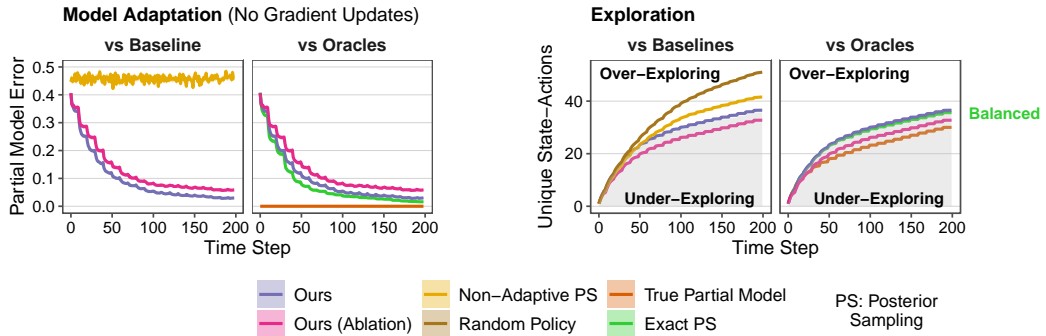

Figure 3: Model adaptation and exploration in held out testing tasks. All lines are averages over held out testing episodes, with the standard error represented by their shaded ribbons (not always visible). **Left:** Our method adapts its partial model inferences with a speed and accuracy close to that of an exact posterior sampling oracle (*Exact PS*). **Right:** Exploration amount is measured by the running number of unique state-actions visited within the partial model space. Our method attains an exploration-exploitation balance close to that of *Exact PS* (green line). Our method's adaptation and exploration happen *in-context*, without gradient-based updates.

presence/absence (1/0 probabilities) of edges in an Alchemy cube. Finally, we measure the amount of exploration as the unique number of state-actions visited in *partial* model space. We choose this metric because it is more representative of how much an agent is gathering useful information for improving its partial model beliefs, compared with state-actions in the full environment.

## 5.2 Reference Point Comparisons

We compare with 2 oracles: *Exact PS* represents an exact implementation of posterior sampling with partial models. Unlike our method, it knows the hypothesis space of 109 possible Alchemy cubes (which includes the testing cubes) and how to perform perfect Bayesian over this space (starting with a uniform prior). It represents an upper bound on adaptation speed and an ideal exploration-exploitation balance within the posterior sampling framework. *True Partial Model* represents posterior sampling when the posterior distribution has already converged to the ground truth partial model, setting a fixed upper bound on rewards (achievable with partial models) and partial model accuracy.

We also compare with 2 baselines: *Non-Adaptive PS* represents posterior sampling with the correct hypothesis space (109 possible Alchemy cubes) but without inference, i.e., it always samples partial models uniformly. *Random Policy* ignores models entirely and uniformly samples actions. They represent reward and exploration amounts without adaptation of partial model beliefs.

## 6 Experiments

**Model adaptation.** Our method adapts its inferred partial model in-context with a speed and accuracy that almost matches an exact implementation of posterior sampling (*Exact PS*) (Figure 3; qualitatively reproduced with two more training seeds in Appendix C). These results suggest our transformer's learned inference over a broader hypothesis space of partial models ($2^{12} = 4096$ cubes) approaches the speed and accuracy of perfect Bayesian inference over a known space of only 109 Alchemy cubes. Within 200 time steps, our method's model error almost reaches the *True Partial Model* oracle's 0 error, even though it has never seen the testing partial models' combinations of edges before. The ablation of our method—which is not performing posterior sampling but rather using the expectation of its posterior—adapts similarly well, but it is slower and less accurate, suggesting that *sampling* leads to better model adaptation. Figure 4 illustrates how our full method has learned to update its posterior beliefs according to new evidence about the presence/absence of edges in a partial model.

**Exploration.** Within the posterior sampling framework, a balanced amount of exploration is represented by the *Exact PS* oracle: *Exact PS* gathers new information only to the extent of helping it find a rewarding policy. In the beginning, it explores more by having a more uniform chance of

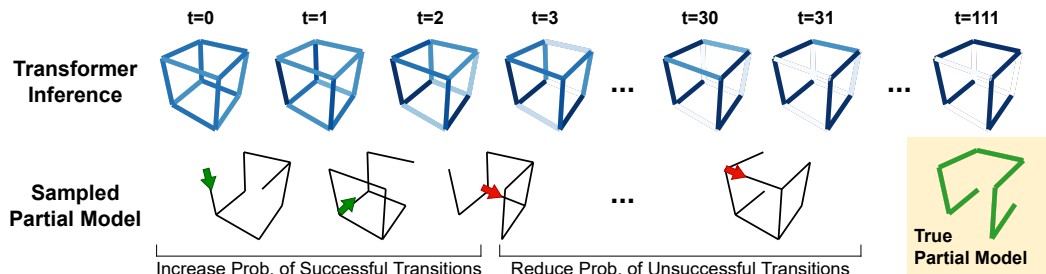

Figure 4: Example of our method's inferences and samples. **Top:** Our transformer's inferences about edge probabilities in a partial model, with darker colors indicating higher probabilities. **Bottom:** Partial models sampled from the transformer's inferences. Our method performs value iteration on these samples to pick an edge to traverse, which is then mapped to a potion to try in the full Alchemy environment. Successful and failed edge transitions are represented by green and red arrows; they are observed in the full environment as a potion transforming the stone or not. Upon receiving such observations in its context (and without gradient updates), our transformer infers higher or lower probabilities for the corresponding edges. Even though it has never seen the true partial model during training, its beliefs come close to the ground truth within only 111 time steps.

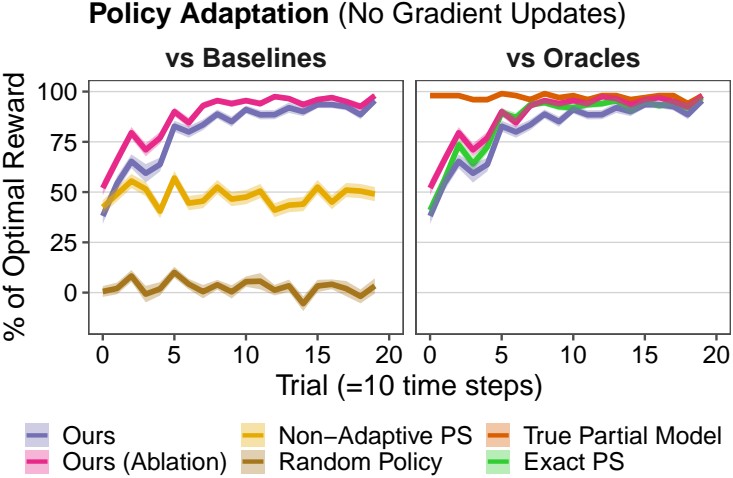

Figure 5: Policy adaptation in held out testing tasks. The lines and shaded ribbons represent the same as in Figure 3. Our method adapts its policy *in-context* to receive around 90% of the highest possible reward, with an adaptation speed approaching that of an exact posterior sampling oracle (*Exact PS*).

sampling different partial models and executing the optimal policy for those samples (Figure 3). As its beliefs become more confident and closer to the true partial model, *Exact PS*'s policy becomes less varied and closer to the true partial model's optimal policy, effectively reducing exploration and focusing more on exploiting its good beliefs. *Random Policy* and *Non-Adaptive PS*, on the other hand, gather information randomly and explore many more unique state-actions in the partial model space, i.e., unique vertex-edges. *True Partial Model* only observes new vertex-edges as it executes the optimal policy for its already-perfect partial model belief; it has no need to explore.

Our results suggest our method strikes a good balance between exploration and exploitation (Figure 3): it explores barely more than *Exact PS*, less than random baselines (*Random Policy* and *Non-Adaptive PS*), and more than the non-exploratory *True Partial Model* oracle. Our ablation has a similar exploration behavior, but it explores *less* than the *Exact PS* oracle and our full method, which can explain why it does not adapt its partial model beliefs as well as our full method. Our ablation explores a smaller number of unique state-actions in the partial model space and likely limits how much context the transformer can leverage to improve its model beliefs. On the other hand, our full method gathers more unique state-actions and thereby enables the same transformer to make better inferences.

**Policy adaptation.**    Our method's in-context policy adaptation also approaches the *Exact PS* oracle's performance in terms of rewards and adaptation speed (Figure 5; qualitatively reproduced with two more training seeds in Appendix C). These results make sense given how well our method adapts its partial model beliefs: as these beliefs improve quickly and accurately, the sampled model is a better description of (a part of) the testing task, meaning value iteration on this model would produce more rewarding policies. Interestingly, our ablation attains slightly higher rewards than our full method in earlier time steps (but are similar at the end of an episode), which is likely due to the differences in how they use information from our transformer.

Whereas our ablation uses all information from our transformer (i.e., the expected value of edges in a partial model) to create a stochastic transition function, our full method throws away some of that information by sampling 1/0 edge probabilities to create a deterministic transition function. Consider the situation when our transformer is certain about its beliefs, i.e., assigns very high or low probabilities to edges. Value iteration on a stochastic transition function would consistently output a policy that traverses high probability edges (i.e., use Alchemy potions) that lead to the most rewarding vertex (i.e., Alchemy stone). In our full method, the sampled transition function may include edges with low (but $> 0$) probabilities or exclude edges with high (but $< 1$) probabilities. For the same transformer beliefs, our full method can produce different transition functions and value iteration would thus produce noisier policies that do not consistently reach the most rewarding vertex (stone).

On the flip side, having a more stable policy implies our ablation explores less effectively than our full method (discussed above and shown in Figure 3), which comes at the cost of worse model adaptation than our full method. We emphasize that effective exploration of a testing task and thereby accurate model adaptation are core focuses of our work. Obtaining a precise model of a testing task is important to gain confidence in the resulting policy.

**Partial models can lead to rewarding policies.**    We had hypothesized that posterior sampling with partial models can lead to rewarding policies even though partial models only capture a part of the full environment's transitions and rewards (see Section 5). Posterior sampling involves dynamic programming (value iteration in our work), which usually assumes it has access to a complete model rather than a partial one. Figure 5 shows that if we have good partial model representations of tasks, as is the case with Alchemy, then posterior sampling with those models can lead to good policies: If an agent's beliefs have already converged to the true partial model of a task, then it can attain almost the same reward as solving the true *complete* model, i.e., almost 100% of the optimal reward (*True Partial Model*). If an agent is still uncertain but has access to a perfect Bayesian inference machine, it can adapt its policy to obtain almost 100% of the optimal reward within 100 time steps (*Exact PS*). If an agent instead has access to an approximate inference process (our transformer), it can adapt its policy to receive around 90% of the optimal reward within 100 time steps (our method).

## 7    RELATED WORK

**Meta-Reinforcement Learning.**    Our work can be viewed as a meta-learning method, defined by an "outer" learning algorithm (supervised learning for updating the weights of our transformer) that learns a "inner" learning algorithm (our transformer's weights implement a learning algorithm that updates a distribution over partial models from context) Hospedales et al. (2022). Essentially, our method "learns to learn" partial models. By combining partial model beliefs with planning (value iteration), our method's inner algorithm becomes a model-based RL algorithm. An inner RL algorithm is a feature of meta-reinforcement learning (meta-RL) methods, and our work is most related to meta-RL methods that have demonstrated *in-context* adaptation. We refer the reader to Hospedales et al.'s (2022) survey for other interesting classes of meta-RL.

Most similar to our method is Rakelly et al.'s (2019) PEARL, which learns an inner algorithm that infers a latent variable from context and samples this variable like in posterior sampling—and Laskin et al.'s (2022) Algorithm Distillation—which learns an inner RL algorithm encoded by transformer weights and uses offline training data. However, these methods (along with others like: Wang et al., 2016; Duan et al., 2016; Zintgraf et al., 2021) are not directly comparable to ours because they adapt policies (in-context) without intermediate model adaptation and planning. They also typically use reinforcement learning or imitation learning for training and, unlike our approach, would not be able to make use of our supervised learning signal.

For meta-RL methods that do adapt models in-context and perform planning, they differ from us because they either do not use neural networks (e.g., Sæmundsson et al., 2018) or usually train networks to predict observations rather than to update a *distribution* over models (e.g., Nagabandi et al., 2019; Weinstein & Botvinick, 2017). We instead focus on training a neural network that can produce a distribution over models for posterior sampling. Another distinction is that we learn small, *partial* models rather than a more typical approach of learning components of a full MDP; our approach enables cheaper planning.

Our training data also differ from many meta-RL works, which typically rely on training data (online or offline) with policy improvement. Our work shows that even data from a *random* policy can be valuable and motivates further investigation into learning from naive policies.

**Abstract MDPs.** Our partial models can also be seen as abstract MDPs containing (1) an abstract state space (i.e., vertices on our partial model graph) that compresses information from the full state space, and (2) transition and reward functions over abstract states (i.e., edges and reward locations on the graph). Both components are features of several works in model-based RL Moerland et al. (2022); Ha & Schmidhuber (2018). Many use a neural network loss that encourages learning abstract states such that they contain enough information for reconstructing the original observations. A slight but interesting difference is that our loss deliberately encourages excluding information from observations (e.g., the abstract state should discard information about whether potions are used up in our environment). Our abstract models need to encode less information than if they were optimized to reconstruct full observations. Notably, the small amount of information that is encoded is quite useful for finding good policies with planning (Figure 5).

The idea of learning abstract models that exclude information from the full MDP but are still useful for planning is also implemented by *value equivalence* methods (e.g., Schrittwieser et al., 2020; Silver et al., 2017; Oh et al., 2017; Farquhar et al., 2018; Tamar et al., 2016). Value equivalence means that abstract models are learned with a loss that encourages accurate predictions of the value function (they do not try to reconstruct the original observations). With respect to the value function, planning with the abstract model should be equivalent to planning in the full MDP. To our knowledge, these methods have yet to tackle our problem of adapting abstract MDPs using context in a new task.

## 8 CONCLUSION

How can an agent generalize *in-context* to new tasks? Our approach is based on posterior sampling Thompson (1933); Strens (2000); Osband et al. (2013) and we approximate components that are often unknown or intractable: Instead of specifying the prior distribution and posterior update in exact Bayesian inference, we use a transformer to *learn* these from training tasks. Instead of computing costly solutions to large MDP models, we consider a hypothesis space of smaller, *partial* models, which are cheaper to solve with dynamic programming. In our version of Symbolic Alchemy, we evaluate how well our method generalizes to held out tasks with novel transition functions. Our method approaches the adaptation speed and exploration-exploitation balance of an exact posterior sampling oracle, and it does so using context alone, i.e., using its history of states and actions rather than gradient updates.

**Limitations and future work.** Here leveraged a supervision signal about how tasks should be represented as partial models, allowing us to instead focus on learning an inference process and testing whether it is reasonable to use partial models in the posterior sampling framework. Our results suggest that partial models that are good representations of tasks may exclude information needed for finding an optimal policy, but planning on them can nevertheless produce very good policies. Future work may be able to borrow ideas from value equivalence methods for *learning* partial model representations without a supervision signal but still ensuring these representations are useful for planning.

### ACKNOWLEDGMENTS

We would like to thank Zita Marinho, Tom Schaul, Jane Wang, Alaa Saade, Michael King and Sam Ritter for helpful discussions and feedback. We are also grateful to Lukas Schäfer and Mhairi Dunion for valuable writing feedback.

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

Table 1: Transformer Hyperparameters

| Hyperparameter | Value |
| --- | --- |
| Embedding size | 32 |
| Num. encoder layers | 4 |
| Num. attention heads | 4 |
| Encoder hidden layer size | 32 |
| Optimizer | Adam (initial learning rate: 1e-4) |

## A   TRANSFORMER ARCHITECTURE AND HYPERPARAMETERS

Our transformer's architecture is based on ideas from several past works that have used an entire data set rather than a single data point as input Kossen et al. (2021); Goyal et al. (2022); Ke et al. (2022); Müller et al. (2022). In our case, we use an entire context matrix rather than a single state-action. Our architecture is set up as follows:

- Input: Context matrix $\mathbf{C}$ of size $T \times N$, where $T$ is the number of time steps in an episode and $N$ is the number of variables in a one-hot encoded state-action $(a_t, s_{t+1})$.
- Embedding: Every $(a_t, s_{t+1})$ row is embedded into a vector of size $H$ using a multi-layer perceptron (MLP).
- Positional encodings are applied over the time step dimension.
- Autoregressive transformer encoder layers: Each layer consists of an autoregressive multi-headed self-attention mechanism, which operates over the time step dimension, and a fully connected feed-forward network.
- Output: The output of the encoder layers is passed to an MLP to make a prediction of size $T \times M$, where $M$ is the length of $\mathbf{e}$ plus the length of $\mathbf{r}$ plus the number of possible vertices. For each time step $t$, the MLP predictions are passed through several sigmoid functions and a softmax function to make the Bernoulli parameters for $\mathbf{e}$ and $\mathbf{r}$, and the categorical distribution of vertices, respectively.

## B   IMPLEMENTING OUR METHOD FOR ALCHEMY

For our simplified Alchemy environment, partial model graphs are constrained to being a cube with 8 vertices and 12 edges (Figure 2), where vertices only represent 3 stone features (color, size, shape) and edges only represent whether it is possible to transform a stone feature with some potion (assuming potions are always full). These partial models' reward function representation $\mathbf{r}$ associates positive reward (=1) only with one vertex, encoding which stone has the highest reward if it is put into the cauldron (all other vertices are associated with zero reward). The transition function representation is a size 12 vector that specifies the probabilities of 12 edges $\mathbf{e} \in [0, 1]^{12}$.

Here context $\mathbf{C}_{1:t}$ is a matrix of an agent's state-action history up to time $t$. States $s$ are represented by vectors with 4 binary indicators for the stone (color, size, shape, whether it is in the cauldron), 6 binary indicators for whether each potion is empty, and a 4-category variable indicating the stone's brightness. Actions $a$ are integers 1-8, representing no-op, putting the stone into the cauldron, and using one of the 6 potions.

We implement our method for our Alchemy setting, meaning our transformer is trained to infer the probability of cube edges $\mathbf{e}$ given a context/history $\mathbf{C}$ of how an agent interacts with stones and potions. Since the reward function is fixed, we do not learn $\mathbf{r}$ with the transformer but instead give it to our value iteration procedure. At testing time, the transformer's inferred edge probabilities imply a policy (via value iteration) that maps vertices to edge traversals, or directly to no-op / "put in cauldron" actions. Finally, edge traversals are mapped back to which potions to try in the full environment.

We generate offline training data in our Alchemy environment: For each of 88 ground truth training cubes (each representing a distinct task), we generate 500 episode-length contexts $\mathbf{C}_{1:T}$ using a random policy, where $T = 200$ is the number of time steps in an episode. Each context is associated

with the supervision signal for the corresponding task $\mathbf{e}_k^*$, i.e., the true cube edges, and with the supervision signal about how each state in the context maps to a vertex $v_t^*$, i.e., the true stone (color, size, shape) in that state. 11 ground truth cubes are used for validation, i.e., selecting the hyperparameters of our transformer, and the final 10 (out of 109 total Alchemy cubes) are used for generating online testing episodes for evaluating our method.

## C  REPRODUCIBILITY ACROSS TRAINING SEEDS

Our results in Figure 3 and 5 are qualitatively reproducible using two other training seeds, as shown in Figures 6-9.

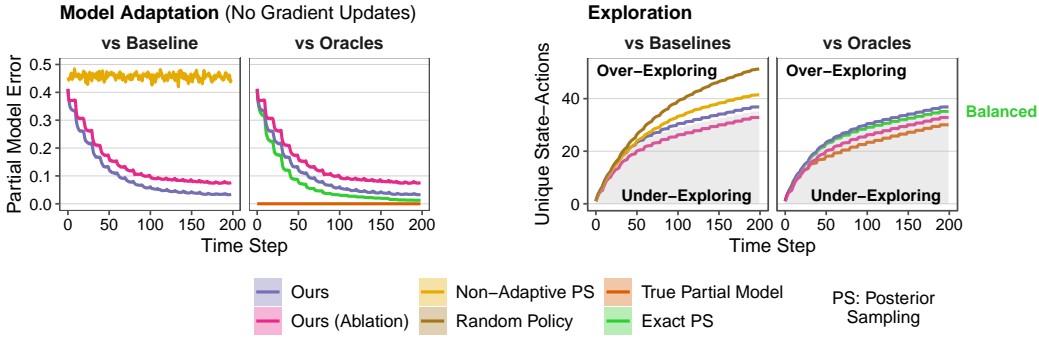

Figure 6: Reproducing Figure 3 with a second training seed.

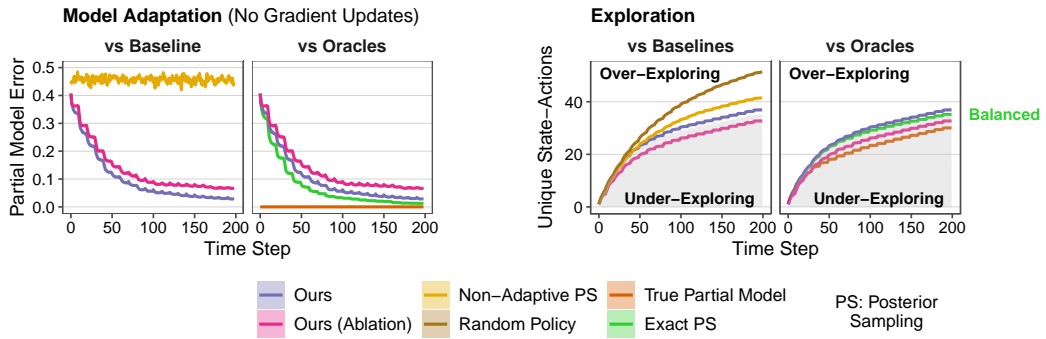

Figure 7: Reproducing Figure 3 with a third training seed.

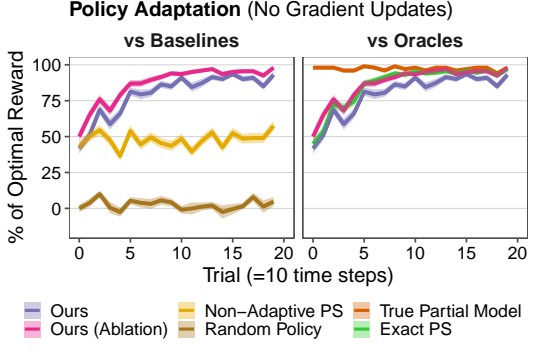

Figure 8: Reproducing Figure 5 with a second training seed.

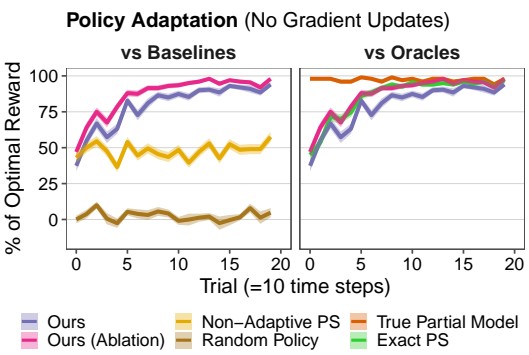

Figure 9: Reproducing Figure 5 with a third training seed.

