# OpenReview forum: "Learning How to Infer Partial MDPs for In-Context Adaptation and Exploration"
_ICLR.cc/2023/Workshop/RRL — RRL 2023 Poster_

### Official Review · Reviewer_g6uu · 2023-02-23
**Good paper, but is it in scope for the workshop?**

**Rating:** 3
**Confidence:** 3

**Review:**

**Review summary**

This paper presents a novel method for test-time adaptation of an RL agent based on training a transformer to model a current belief over abstract, partial world models. The method makes sense, is demonstrated clearly on toy data, and the paper is of a high quality.

Despite the strong assumptions, which limit the method to toy settings for now, I believe that this paper is certainly good enough to be accepted, with one catch: the conference organizers need to decide if this paper is in scope for the workshop, as it is not really about the reincarnation of prior knowledge or computation.

**Paper summary**
- The authors study the problem of in-context adaptation: solving new tasks with an agent solely based on real-time information, without updating its weights (as in many meta-RL methods, for instance).
- They propose a solution based on approximate posterior inference over partial models. Partial models are abstract, simplified MDPs represented as vectors that specify transition and reward models. A transformer is trained to learn a belief over these partial models conditional on past experience on the current task.
- To make this problem tractable, the authors make two key assumptions. They focus on problems where 1) training tasks are labeled with the relevant partial model and 2) the states in the partial model directly correspond to actions in the original environment. With these assumptions, training the transformer is a supervised learning problem, and the optimal policy in the reduced MDP can be transferred to the original MDP easily. The challenging problem of how to learn the state and action representations from interactions with the raw environment are not addressed.
- At test time, the authors iterate between sampling a partial model from the transformer and finding the optimal policy under this partial model through value iteration.
- The method is demonstrated on a toy environment, where it approaches oracle performance.

**Quality**
- Within the assumptions, the method is sound.
- The demonstration in a toy experiment is convincing.

**Significance**
- The paper works on the interesting and relevant problem of test-time adaptation to new tasks.
- However, the two key assumptions of 1) partial model labels and 2) a direct correspondence between partial model states and actions in the full MDP limit the impact of this method as is. In its current form, I cannot think of realistic problems that the algorithm could solve (I'd love to hear some examples from the authors if they have any). The authors are transparent about this limitation and I believe that there is still enough interesting content in this paper to make it a good research contribution.
- As far as I can tell (I am not an expert), the method is novel.
- The fit for the workshop is a little bit less clear. In which sense is the method reusing prior data or computations? I would like to leave it up to the organizers to decide whether this paper is in scope.

**Clarity**
- Overall, the paper is very well written.
- There are two topics which (in my opinion) deserve some further discussion, see sections below.
- Visualizaations and plots are of a high quality.
- Related work is discussed in depth.

**Discussion of the in-context adaptation setting**
- I am curious whether there is a way of defining "in-context adaptation" purely through the *behaviour* of the agent, without replying on its inner workings (like whether it performs gradient updates). Can there be a decision rule that just looks at the sequence of states, actions, and rewards that result from the agent-environment interaction, and then outputs whether the agent performs in-context adaptation?
- If I understand the intention of the authors correctly, there are two criteria that make for "in-context adaptation": the first is the fact that the agent is stateless after the end of training. It cannot share experiences collected at test time except through the explicit conditioning. The second is the fact that the agent is computationally efficient. This forbids gradient updates as part of the decision process and forces the agent to stick to a single forward pass.
- If my interpretation is correct, I would politely suggest to the authors to update the description in Section 2 a bit. Currently, they write "At testing time, models and policies should improve as a function of conditioning on longer contexts". It is pretty clear what they have in mind. However, being a bit nitpicky, this phrasing does not really rule out agents that perform gradient steps "internally" on the context.

**Discussion of the partial MDPs**
- The partial MDPs here have a restricted transition model. If I understand correctly, each action corresponds to a single edge (so a single target state in the reduced MDP). There is thus no way for stochastic transitions in which a single action could let the agent end up in multiple different states. Is this correct? If yes, maybe add a discussion about the usefulness of this setting?
- Another assumption of the partial MDP is that of binary rewards. This also deserves a bit more discussion.

**Minor comments and questions**
- Line 36, right column: In which sense is approximate inference really an antonym to Bayesian inference? To me, this sentence would make more sense if it said "We use approximate rather than exact Bayesian inference" and so on
- Line 98-99: maybe explicitly add the dependence on the task k to the reward function and transition dynamics?
- Line 110: why does the context not include the rewards achieved by the agent?
- Line 127: what is this categorical distribution over vertices used for? This is mentioned in Section 4, but would be good to already explain here.

---

### Official Review · Reviewer_uH6X · 2023-02-25

**Rating:** 2
**Confidence:** 3

**Review:**

The authors tackle the challenging problem of in-context adaptation and exploration without using gradient-based updates. They approach this challenge by training partial models in a supervised manner using a Transformer architecture that are used as the hypothesis space in the inference process. Their results show that using the pre-trained partial models facilitate adaptation and lead to good policies even excluding relevant information from the environment.

Strengths:

1. The problem of in-context adaptation and exploration is challenging and significant advances in the methods used for this problem are needed in the literature.
2. The idea of learning general knowledge in the form of an inference process over partial models is very interesting, and representing the partial models as graphs that can be approximated using neural networks (Transformers or Graph Neural Networks) is an interesting and promising approach.
3. The results show that the trained partial models can lead to rewarding policies that can be effectively used with posterior sampling.

Weaknesses:

1. Even the problem is challenging and the idea is interesting, the clarity of the paper could be improved with more Figures and Diagrams that helps the reader to understand the available training data, which policies are trained, how the trained policies are used, and how the posterior sampling works.
2. The partial models are trained in a supervised manner using data from a random policy (which is ok depending on the environment). In the conclusion, the authors discuss that a future work is to learn partial model representation without supervision signals, but the reviewer thinks that an in-depth discussion comparing the partial models with other model-based methodologies, intrinsic rewards, and reward prediction, might be needed.
3. The paper might be at the edge of the workshop’s scope. There is not an in-depth discussion regarding the reuse of data collected by previous policies that can be used to train the partial models or the reuse of world models.

Overall, the reviewer’s opinion is that the work has merit and tackle an important and challenging problem. Even the paper tackles a problem relevant for the workshop theme, the reviewer thinks that additional discussion might improve the clarity of the paper and its contributions.